# IMU2CLIP: Language-grounded Motion Sensor Translation with Multimodal Contrastive Learning

**Seungwhan Moon**[*]   **Andrea Madotto**[*]   **Zhaojiang Lin**   **Aparajita Saraf**
**Amy Bearman**   **Babak Damavandi**
Meta Reality Labs & FAIR, Meta

## Abstract

We present IMU2CLIP, a novel pre-training approach to align Inertial Measurement Unit (IMU) motion sensor recordings with text and video, by projecting them into the joint representation space of Contrastive Language-Image Pre-training (CLIP). The proposed approach allows IMU2CLIP to *translate* human motions (as measured by IMU sensors) into their corresponding textual descriptions and videos – while preserving the *transitivity* across these modalities. We introduce several new IMU-based Wearable AI applications such as motion-based media search, or an LM-based multimodal reasoning with motion sensor data – all using text as the grounding platform. In addition, we show that IMU2CLIP significantly improves downstream performances when fine-tuned for each application, demonstrating its universal usage as a new pre-trained resource. Our code and models will be released publicly.

## 1 Introduction

With the growing popularity of smart glasses or new-generation wearable devices, *first-person* or *egocentric* videos have recently become prevalent (Grauman et al., 2022; Damen et al., 2021; Lv et al., 2022). These egocentric videos are often accompanied by the parallel head-mounted IMU sensor readings, which record devices' linear and rotational movements and accelerations.

Given its low power consumption level, IMU is regarded as an important modality for powering various *always-on* on-device models that require understanding of device wearer's movement patterns (*e.g.* exercise / activity recognition for health applications) – which would not work with more battery-consuming camera sensors. The previous works on IMU modeling typically focus on the purpose-built datasets with manual annotations (Jiang et al., 2022; Chen et al., 2021), which are limited in their

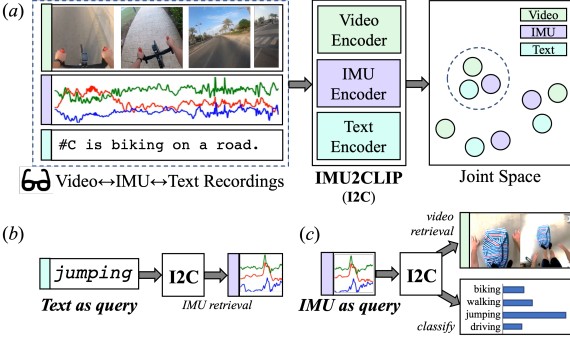

Figure 1: Illustration of IMU2CLIP (I2C): (*a*) The model aligns the parallel Video↔IMU↔Text data in the joint space. Once trained, IMU2CLIP is used as a retriever for both (*b*) IMU and (*c*) videos, or as a text-based zeroshot classifier for downstream applications.

scale. Consequently, the utilization of IMU models in real-world scenarios has been confined to a relatively small number of use cases.

On the contrary, for the modalities that are widely studied (*e.g.* text, video), there are vast large-scale resources such as BERT (Devlin et al., 2018) and GPT (Radford et al., 2018) for text, or CLIP4Clip (Luo et al., 2021) for videos. These powerful pre-trained resources have driven the development of many application-oriented models, showing significant improvements when fine-tuned for each respective task (Dodge et al., 2020). To our knowledge, however, the study on the equivalent resources for IMU signals has been lacking.

Inspired by the recent works studied for other modalities, we present IMU2CLIP, a new approach to pre-train an IMU encoder by aligning the parallel Video↔IMU↔Text data. Specifically, we propose to use CLIP (Radford et al., 2021a), which contains the video and language models pre-trained on the large image-text parallel data. The IMU encoder thus learns semantic representations of various scenes transferred from other modalities.

Note that anchoring onto the joint text-vision space essentially allows for ***translation*** across these modalities. This new model capability opens up

---

[*]Joint First Authors. {shanemoon,andreamad8}@meta.com

novel applications in the real-world Wearable AI, such as IMU-based media search and LM-based multimodal reasoning (Fig. 1) – serving as efficient low-power alternatives to a more power-consuming approach of video-based media retrieval, which can operate under the strict on-device hardware constraints of wearable devices. In our evaluation, we provide an in-depth analysis on these newly proposed applications, as well as its efficacy on the downstream applications when fine-tuned.

## 2 Related Work

**Contrastive Learning** is as an efficient self-supervised framework applied across multiple domains, which learns similar/dissimilar representations from paired data. For instance, SimCLR (Chen et al., 2020) is a unimodal application of contrastive learning in the data augmentation setting, which proposes to learn a vision encoder given a set of perturbed images. As an example in multimodal settings, Contrastive Language–Image Pre-training (CLIP) (Radford et al., 2021a) learns visual representations from natural language supervision using image and text pairs, achieving competitive results in *e.g.* zero-shot image classification, image retrieval via text, and image/caption generation. Similarly, WAV2CLIP (Wu et al., 2022) proposes to learn audio representation by distilling it from CLIP. We extend this line of work to a unique multimodal setting that utilizes IMU signals, which is specific to a new generation of devices (such as smart glasses) that are equipped with such sensors.

**Pre-training Resources**: There are numerous pre-trained resources for well-studied modalities such as text or image. Popular language models (LM) include BERT (Devlin et al., 2018), GPT-2 (Radford et al., 2019), and GPT-3(Floridi and Chiriatti, 2020), which typically use self-superivsion techniques such as next-word predictions or masked token predictions, thus not requiring any explicit task labels. Studies report that these pre-trained resources achieve competitive zero-shot performance (Radford et al., 2021a), and when fine-tuned, often outperform fully supervised models on several downstream tasks (Dodge et al., 2020).

To our knowledge, the equivalent resource for IMU is not made publicly available. We perform large-scale pre-training for the unique wearble sensor signals dataset, and show that such pre-training significantly improves downstream performances.

**Egocentric Datasets**: We focus on egocentric (first-person) datasets, for understanding of users' activities from head-mounted devices. These include Ego4D (Grauman et al., 2022), Epic-Kitchens (Damen et al., 2021), and Aria (Lv et al., 2022). Using these datasets, we propose various sub-tasks that can effectively evaluate diverse capabilities of IMU2CLIP, and demonstrate the feasibility of future applications. In addition, we implement a universal multimodal dataloader to allow for easy cross-modality and cross-domain studies.

**IMU Modeling**: IMU signals are used in various motion recognition tasks, such as pose estimation and walking speed estimation (Zihajehzadeh and Park, 2017). Various architectures (Kim et al., 2021; Ashry et al., 2020) have been explored for modeling IMU in downstream tasks, including Transformer-CNN based IMU models (Jiang et al., 2022). Our work proposes a new IMU model architecture, and conducts ablation studies over other models above. Different from prior works that train IMU models for specific tasks, however, our work focuses on learning *multimodal* IMU representations by aligning IMU with text and image, which can enable a wider set of downstream applications.

## 3 Methods

In a nutshell, we train an IMU encoder such that the IMU representation of a given clip resembles the representation of its corresponding textual descriptions (narrations), or corresponding video frames. Fig. 5 illustrates the overall approach.

**Cross-modal Contrastive Learning.** We consider a batch of $B$ ground-truth $\underline{I}MU \leftrightarrow \underline{V}ideo \leftrightarrow \underline{T}ext$ parallel windows: $\{(\mathbf{i}_1, \mathbf{v}_1, \mathbf{t}_1), ..., (\mathbf{i}_B, \mathbf{v}_B, \mathbf{t}_B)\}$, where the embeddings of each modality lies on the unit hypersphere $S^D$. Since they are unit-normalized, the similarity can be calculated as their inner products: $\text{sim}(\mathbf{i}_i, \mathbf{v}_j) = \langle \mathbf{i}_i, \mathbf{v}_j \rangle$ and $\text{sim}(\mathbf{i}_i, \mathbf{t}_j) = \langle \mathbf{i}_i, \mathbf{t}_j \rangle$. We then train three flavors of IMU2CLIP: (a) aligning IMU↔Text, (b) IMU↔Video, and (c) IMU↔Video↔Text. Specifically, we project the IMU representations into the joint CLIP space (Radford et al., 2021a) to leverage the visual and textual knowledge already encoded in CLIP. Similar to (Luo et al., 2021; Radford et al., 2021a), we propose to minimize the symmetric cross-modal contrastive loss (ablations on loss choices in Appendix). For the IMU↔Text alignment, we use the sum of IMU-to-Text ($\mathbf{i}2\mathbf{t}$) and Text-to-IMU ($\mathbf{t}2\mathbf{i}$) cross-entropy losses:

$$\mathcal{L}_{\mathbf{i}2\mathbf{t}} = -\frac{1}{B} \sum_{i=1}^{B} \log \frac{\exp(\text{sim}(\mathbf{i}_i, \mathbf{t}_i))^{1/\gamma}}{\sum_{k=1}^{B} \exp(\text{sim}(\mathbf{i}_i, \mathbf{t}_k))^{1/\gamma}}$$

| Dataset | Statistics | Tra. | Val. | Tst. |
|---------|-----------|------|------|------|
| **Ego4D** | # Media files | 1444 | 161 | 688 |
| | Total Media Durations | 540h | 60h | 265h |
| | # IMU↔Text/Video Pairs | 528K | 68K | 266K |
| | # Windows for Labels | 1552 | 760 | 241 |
| **Aria** | # Media files | 747 | 259 | 277 |
| | Total Media Durations | 138h | 43h | 51h |
| | # IMU↔Video Pairs | 496K | 157K | 184K |
| | # Windows for Labels | 25K | 138K | 162K |

Table 1: Dataset Statistics for Ego4D and Aria.

where $\mathcal{L}_{\mathbf{i}\leftrightarrow\mathbf{t}} = 1/2\,(\mathcal{L}_{\mathbf{i2t}} + \mathcal{L}_{\mathbf{t2i}})$, with $\mathcal{L}_{\mathbf{t2i}}$ defined symmetrically. The loss for IMU↔Video alignment ($\mathcal{L}_{\mathbf{i}\leftrightarrow\mathbf{v}}$) can be defined similarly, and consequently $\mathcal{L}_{\mathbf{i}\leftrightarrow\mathbf{v}\leftrightarrow\mathbf{t}} = \mathcal{L}_{\mathbf{i}\leftrightarrow\mathbf{v}} + \mathcal{L}_{\mathbf{i}\leftrightarrow\mathbf{t}}$.

**IMU Encoder Architectures.** We propose an architecture with a stack of 1D-CNNs and a RNN, which performed the best in our ablation studies (See Appendix. A.2). First, we perform Group-Norm to normalize the Accelerometer (3D) and the Gyroscope (3D) signals independently. We then add a stack of $N = 3$ 1D-CNNs, a Max Pooling layer (kernel size 5), and another GroupNorm to normalize the output features. We use GRU to combine the CNN outputs as final representations.

## 4 Experiments

**Dataset.** We use Ego4D (Grauman et al., 2022) and Aria (Lv et al., 2022) as the main datasets for the experiments below, both of which feature large-scale parallel video and IMU signals. For Ego4D, a subset of the clips are also annotated with text narrations. We split the data into train, validation, and test sets (split by video IDs). The statistics of the datasets are provided in Table 1.

### 4.1 Tasks

Note that the proposed pre-training approach enforces the alignment of the IMU, text, and video representations, which allows for new and unique cross-modal applications. We propose the following real-world downstream applications as the novel tasks to evaluate the IMU encoders.

**Task 1. Media Search via Text→IMU Retrieval**, where the goal is to retrieve a window of IMU signals given free-form textual queries. Once IMU signals are retrieved, we can also retrieve their corresponding videos, thus allowing for a new and power-efficient way of performing media retrieval or online action detection. We evaluate on the held-out test set (Recall@$k$ and Mean Reciprocal Rank (MRR)), using the text narrations as queries and the IMU signals as the retrieval pool.

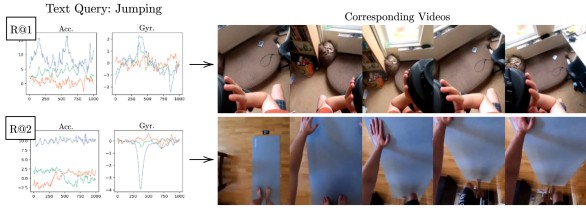

Figure 2: Media Search via Text→IMU Retrieval. Given a free-form text query (*e.g.* "*jumping*"), (Left): IMU2CLIP's predictions of the semantically closest IMU signals from Ego4D (top-2). (Right): corresponding gold-parallel videos (as a reference). Retrieved results match the semantics of input queries.

**Task 2. IMU-based Activity Recognition**, where the goal is to predict a discrete motion-based activity label (*e.g.* hiking, running, walking, biking) given a window of IMU signals, measured via F1.

**Task 3. Question Answering on Sensor Logs**, where the goal is to respond to users' memory recall queries with natural language, based on the logs from ambient sensors (*e.g.* IMU, audio), as an exploratory task. We leave the quantitative evaluation of this task as future work.

### 4.2 Results

Table 2 shows the IMU↔Text and IMU↔Video retrieval performance on the Ego4D test set, of IMU2CLIP trained via different combinations of modalities. Results on Aria are in Appendix A.3.

**IMU-based media search with textual queries**: The Text→IMU column in Table 2 shows performances on Task 1. Note that the CLIP embeddings already exhibit the Video↔Text alignment, and thus IMU2CLIP trained using IMU↔Video achieves a competitive zeroshot performance for IMU↔Text retrieval as well. When text narrations are used for pre-training, the model learns IMU representations that are better aligned with language, and thus achieves an even higher recall performance. See Figure 2 for visualizations[1].

To help contextualize the performances in Table 2, we also show (as a reference) the Text↔Video retrieval performance of the CLIP video encoder. The narrow margins (*e.g.* MRR=0.143 for Text→IMU *vs.* MRR=0.168 for Text→Video) show that the IMU encoder could serve as a power-efficient alternative for a video encoder in many applications, where the memory and power usage is restricted (*e.g.* on wearable devices).

**Fine-tuned IMU2CLIP significantly outper-**

---

[1]For better readability, we also provide the animated GIF visualizations in the supplementary materials.

| Train Modalities | | | IMU → Text | | | | Text → IMU | | | | IMU → Video | | | |
|---|---|---|---|---|---|---|---|---|---|---|---|---|---|---|
| IMU | Video | Text | R@1 | R@10 | R@50 | MRR | R@1 | R@10 | R@50 | MRR | R@1 | R@10 | R@50 | MRR |
| ✓ | ✓ |  | 4.86 | 18.75 | 48.26 | 0.104 | 4.17 | 15.62 | 43.06 | 0.084 | 9.06 | 43.13 | 78.75 | 0.2011 |
| ✓ |  | ✓ | 5.21 | 25.00 | 60.42 | 0.123 | 7.29 | 28.82 | 60.07 | 0.143 | 3.75 | 25.94 | 62.81 | 0.105 |
| ✓ | ✓ | ✓ | 4.52 | 22.91 | 56.60 | 0.118 | 5.90 | 22.92 | 56.60 | 0.139 | 8.75 | 40.63 | 73.44 | 0.183 |
|  | | | **(Video → Text)** | | | | **(Text → Video)** | | | | - | | | |
| (CLIP) | ✓ | ✓ | 6.94 | 32.29 | 64.24 | 0.150 | 8.33 | 33.68 | 65.28 | 0.168 | | | | |

Table 2: Text↔IMU and Video↔IMU retrieval performances of the pre-trained IMU2CLIP models on Ego4D, with different modalities used for training. The last row shows the video retrieval performance of OpenAI's CLIP model on the same test set.

| Models | | Ego4D | | Aria | |
|---|---|---|---|---|---|
| | | F1 | Acc. | F1 | Acc. |
| Vanilla IMU Encoder | | 23.23 | 49.92 | 56.35 | 76.11 |
| IMU2CLIP ($\mathbf{i} \leftrightarrow \mathbf{v}$) | + Zeroshot | 19.39 | 23.08 | 18.46 | 21.52 |
| | + Probing | 40.55 | 61.46 | **62.52** | **83.54** |
| | + Fine tuning | 43.07 | **65.87** | 61.77 | 82.31 |
| IMU2CLIP ($\mathbf{i} \leftrightarrow \mathbf{t}$) | + Zeroshot | 31.89 | 36.38 | | |
| | + Probing | 45.12 | 58.01 | N/A | |
| | + Fine tuning | **45.15** | 63.14 | | |

Table 3: IMU-based activity recognition on Ego4D and Aria datasets, comparing the randomly initialized model (vanilla) and the pre-trained IMU2CLIP models, with IMU↔Video, and IMU↔Text. **Bold** denotes the best performance for each metric: F1 and Accuracy (Acc).

```
[15:44] Motion: no activity, Audio: no activity
[15:45] Motion: walking, Audio: TV noise
[15:50] Motion: sits down, Audio: watching TV
[15:52] Motion: look around, Audio: kitchen sounds
[16:03] Motion: walking, Audio: no activity
Question: how long did I sit down and focus on watching TV?
Answer: From 15:50 to 16:03, for a total of 13 minutes.
```

Figure 3: Demonstration of LM-based multimodal reasoning via IMU2CLIP, using two ambient sensors: IMU and audio. Given the sensor logs (translated in text) and the question, LM generates responses grounded on the multimodal context (More examples in the Suppl.).

**forms the vanilla model with the same architecture trained from scratch, on downstream tasks.** Table 3 shows the activity recognition results on Ego4D and Aria. For all experiments, we use the same IMU architecture (Stacked CNN). For zeroshot experiments, we encode the surface names of each activity (*e.g.* hiking) with the CLIP text encoder, and use the nearest-neighbor classifier on the projected IMU embeddings (thus without any supervision labels). Probing adds a linear layer on top of the IMU encoder while keeping its parameters frozen, and for fine-tuning we allow all parameters to be trainable. The consistent improvements in the fine-tuning performances (*e.g.* ∼**16 points** absolute improvement in accuracy for Ego4D, comparing the randomly initialized model *vs.* fine-tuned model) show that IMU2CLIP can learn high quality representations for IMU signals.

Comparing the pre-trained models trained via various combinations of modalities again shows that IMU2CLIP preserves the transitivity among modalities (video ↔ text ↔ IMU).

**Qualitative Analysis: Multimodal Reasoning with Ambient Sensors.** Further exploring the benefit of IMU2CLIP that translates sensor signals into text, we present the following demo (Figure 3). Specifically, we run IMU2CLIP (and an equivalent model trained for audio) as zeroshot tagging models on Ego4D, to predict textual descriptions of the ambient sensor readings (IMU+Audio). Using text as a grounding platform, we condition a large LM (GPT-3 (Floridi and Chiriatti, 2020)) on the sensor logs to answer summarization questions (*e.g.* "*What can you tell me about the user activity?*") or memory recall prompts (*e.g.* "*When did I start biking?*") The LM then generates a response via causal language inference, demonstrating zeroshot reasoning capabilities. Unlike the similar approach by (Zeng et al., 2022), the proposed approach does not rely on video signals at all – which incur much higher power consumption – thus operating better under real-world *on-device* constraints.

## 5 Conclusions

With the growing popularity of wearable devices of diverse form factors (*e.g.* smart glasses), it is imperative to study ambient motion sensors such as IMU. We thus make the following **contributions**: 1) We propose a large-scale pre-training method for IMU↔Text, and release the pre-trained multimodal encoders for future research. (2) We provide an in-depth empirical analysis for both upstream and fine-tuning tasks. (3) Most importantly, we demonstrate the feasibility of many multimodal NLP applications for ambient sensors via their translation to text, which can spur future research.

## 6 Limitations

We discuss the current limitations of our work:

(1) While we used the two of the largest egocentric multimodal datasets (Ego4d (Grauman et al., 2022) and Aria (Lv et al., 2022)) that are publicly available, the transferability of this work to an unseen data (*e.g.* from other publicly available wearable devices, or of different activity domains) remains unanswered. We do note, however, that the hardware specifications of most IMU sensors and video sensors are standardized across the industry, hence posing limited risks.

(2) Our demonstration of the multimodal reasoning model (Figure 3) is currently at a qualitative level, due to the lack of a benchmark dataset on this newly proposed task. While our analysis shows promising results, a quantitative evaluation is required to further determine the feasibility of this task and to track progress in the field. However, we emphasize that the purpose of this work is to *explore* potential applications that the proposed approach brings forth, and to inspire future research in a fast-growing application domain that uses smart glasses.

## 7 Ethics and Broader Impacts

We hereby acknowledge that all of the co-authors of this work are aware of the provided *ACL Code of Ethics* and honor the code of conduct. We state the ethical considerations and the potential impact to the community as follows.

**Datasets.** The datasets we use to train our model (Ego4D, Aria) are the commissioned data that are released publicly. Both of the datasets note that the consent forms and/or release forms are collected for all videos, and that the majority of videos and data are de-identified pre-release.

In our code release, we provide the preprocessing scripts for the 3rd party datasets above (along with the links to download these datasets). We note that the pre-processed data (ready for training) would not contain any of the meta information that is not needed for the purpose of this work – such as date or location of the recordings.

**Applications.** The models we train and the tasks that we propose are inherently made for *on-device* use cases – where the model operates on the data that are already publicly released (such as the public datasets above) or with an explicit consent, thus posing minimal risks.

**Techniques.** We train the proposed IMU2CLIP model leveraging other large-scale pretrained language and multimodal models (*e.g.* CLIP (Radford et al., 2021a), OPT (Zhang et al., 2022), GPT-3 (Floridi and Chiriatti, 2020)), and thus any bias or contextual information from these models might be carried over to our model as well (de Vassimon Manela et al., 2021). Given the nature of the datasets we use, however, we do not anticipate that our models would produce any harmful outputs, especially towards vulnerable populations, after the models are trained on our tasks.

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

## A Additional Experiments

### A.1 Ablation Studies over Loss Functions

We survey the performance of the modality alignment via different loss functions: InfoNCE contrastive loss (used throughout our main experiments), Triplet loss with random sampling for negative pairs, and Mean-Squared Error loss (MSE).

### A.2 Ablation Studies on IMU Architecture

We survey the performance of IMU encoders with varying architectures (see Section 2 for the list of IMU baselines in the literature), and use the best architecture throughout the rest of the experiments.

As seen in Table 5, our proposed 1D-CNN-RNN stacked model outperforms other baselines in the main target task (IMU to Text retrieval). See Section 3 for the detailed description of the model.

### A.3 IMU↔Video Retrieval

We propose an auxiliary task of **Video Retrieval based on IMU**, specifically targeted for Aria data, which does not have text narrations annotated. The goal for the task is to retrieve videos based on IMU signals, allowing for an intuitive way of analyzing motion signals data. We measure the performance on the held-out test set, using the IMU signals as queries and the videos as the retrieval target.

**We can search for videos, given IMU recordings**. Table 6 shows the performances on the IMU↔Video retrieval task on Ego4D and Aria datasets. We observe higher recall performances in general for both datasets, showing that the IMU signals and videos have a higher compatibility. When the model is trained on all three modalities, we observe competitive results across all tasks, while the best performances are from the bi-modal models aligned with each respective task.

Fig. 4 shows illustrative examples, which demonstrate that the videos retrieved based on IMU are visually and semantically similar to the gold video.

## B Implementation Details

### B.1 Training Details

**Model Freezing**. To preserve the text-vision alignment that CLIP already exhibits, we freeze the parameters of the image and text CLIP encoders during training. See Fig. 5 for an illustration.

**Memory Constraints**. To expedite the training,

| Loss Function | MRR | R50 |
|---|---|---|
| InfoNCE loss | **0.123** | **60.42** |
| Triplet Loss | 0.118 | 59.53 |
| MSE Loss | 0.101 | 48.55 |

Table 4: Ablation studies with varying loss functions on the IMU→Text Retrieval task. Model: Stacked 1D-CNN-RNN. **Bold** denotes the best performance.

| IMU Encoder Architecture | MRR | R50 |
|---|---|---|
| Stacked 1D-CNN-RNN (**proposed**) | **0.123** | **60.42** |
| 1xCNN+Attention+RNN | 0.112 | 57.18 |
| Patch Transformer | 0.109 | 54.15 |
| Patch Bi-RNN | 0.112 | 56.21 |
| Patch RNN | 0.104 | 50.83 |

Table 5: Ablation studies for varying IMU encoder architectures on the IMU→Text Retrieval task. **Bold** denotes the best performance.

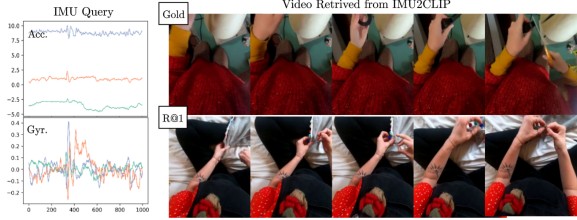

Figure 4: Video Retrieval based on IMU (IMU→Video). (Left): IMU signals and (Top): their corresponding ground-truth video. (Bottom): IMU2CLIP 's model prediction of their corresponding video from the Ego4D test set (top-1), given the IMU signals. It can be seen that the videos retrieved based on IMU are visually and semantically similar to the gold video.

we pre-process each media to have equal-sized parallel windows (IMU ↔ Video ↔ Text). The data module retrieves the parallel data of a requested window size at a given timestamp, and caches them for faster training. In addition, to accommodate the memory constraints, we pool the negative samples within the same batch (randomly shuffled), reducing the load on each GPU.

### B.2 Hyperparameters

Table 7a and 7b report the hyper-parameters used in this work for model training and their search bounds, respectively. We optimize the parameters with Adagrad (Duchi et al., 2011) with epsilon $10^{-8}$, and decay 0.1.

| Train Modalities | | | IMU → Video | | | | Video → IMU | | | |
|---|---|---|---|---|---|---|---|---|---|---|
| IMU | Video | Text | R@1 | R@10 | R@50 | MRR | R@1 | R@10 | R@50 | MRR |
| | | | **Ego4D** | | | | | | | |
| ✓ | ✓ | | 9.06 | 43.13 | 78.75 | 0.2011 | 12.19 | 45.31 | 80.00 | 0.226 |
| ✓ | | ✓ | 3.75 | 25.94 | 62.81 | 0.105 | 3.75 | 24.06 | 56.88 | 0.098 |
| ✓ | ✓ | ✓ | 8.75 | 40.63 | 73.44 | 0.183 | 11.56 | 42.19 | 75.94 | 0.213 |
| | | | **Aria** | | | | | | | |
| ✓ | ✓ | | 7.58 | 40.62 | 83.92 | 0.181 | 7.58 | 45.08 | 83.92 | 0.194 |

Table 6: Video↔IMU retrieval performances of the pre-trained IMU2CLIP models on Ego4D (top) and Aria (bottom), with different modalities used for training. The last row shows the video retrieval performance of OpenAI's CLIP model on the same test set.

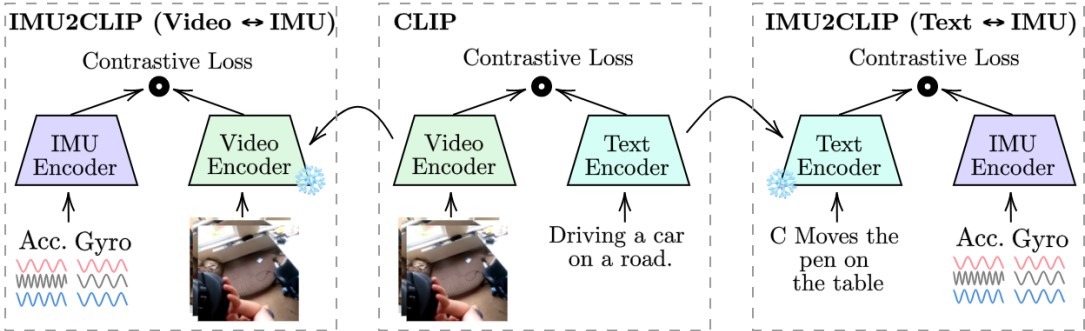

Figure 5: Illustration of the proposed multimodal contrastive learning for IMU2CLIP. CLIP (Radford et al., 2021b) is used to align IMU↔Video (left), and IMU↔Text (right). ❄: the parameters of the encoder are frozen during training.

| Models | Batch Size | Initial LR | # Epochs | Gradient Accumulation Steps | # Params |
|---|---|---|---|---|---|
| Stacked 1D-CNN-RNN (IMU+Text) | 16 | $2 \times 10^{-4}$ | 10 | 1 | 1.7M |
| Stacked 1D-CNN-RNN (IMU+Video) | 16 | $5 \times 10^{-4}$ | 16 | 1 | 1.7M |
| Stacked 1D-CNN-RNN (IMU+Text+Video) | 16 | $5 \times 10^{-4}$ | 16 | 1 | 1.7M |

(a) Hyperparameters for Pre-training

| Type | Batch Size | Initial LR | # Training Epochs | Gradient Accumulation Steps |
|---|---|---|---|---|
| **Bound (lower–upper)** | 4–32 | $5 \times 10^{-5}$–$5 \times 10^{-3}$ | 6–10 | 1–1 |
| **Number of Trials** | 5 | 5 | 5 | 1 |

(b) Search Bounds

Table 7: **(a) Hyperparameters in this work:** *Initial LR* denotes the initial learning rate. All the models are trained with Adagrad optimizers (Duchi et al., 2011). We include the number of learnable parameters of each model in the column: *# params*. **(b) Search bounds** for the hyperparameters of all the models.

### B.3 Code Base & Hardware

The implementations of the transformer-based models are extended from the HuggingFace[2] code base (Wolf et al., 2020) and other cited authors' released code-bases. Our entire code-base is implemented in PyTorch (Paszke et al., 2019). All models in this work are trained on a varying number of Nvidia A100 [3] GPUs (1-8 depending on the availability) on a Ubuntu 20.04.2 operating system.

### C Supplementary Materials

We also provide the animated GIF visualizations (as a zipped file) for all of the illustrative tasks mentioned in this manuscript, as part of the uploaded supplementary materials.

---

[2] https://github.com/huggingface/transformers
[3] https://www.nvidia.com/en-us/data-center/a100/