# OpenReview forum: "IMU2CLIP: Language-grounded Motion Sensor Translation with Multimodal Contrastive Learning"
_EMNLP/2023/Conference — EMNLP 2023 Findings_

### Official Review · Reviewer_b6Gx · 2023-07-27

**Soundness:** 3

**Excitement:**

3: Ambivalent: It has merits (e.g., it reports state-of-the-art results, the idea is nice), but there are key weaknesses (e.g., it describes incremental work), and it can significantly benefit from another round of revision. However, I won't object to accepting it if my co-reviewers champion it.

**Paper Topic And Main Contributions:**

This paper introduces a pre-training approach to align 3 modalities of data including IMU (Inertial Measurement Unit, used for capturing movements and orientations) and two common ones of video and text. Video and text inputs are encoded with pre-trained CLIP model, while the IMU input is encoded by customized stacks of CNN and RNN layers. All inputs, after being encoded, are unified by two contrastive losses conjugating the pair of IMU and video, as well as IMU and text. The authors evaluated the trained model on two main tasks (1) retrieve IMU from text (2) IMU to actions, and results show improvement from incorporating both video and text modality in the pre-training.

**Questions For The Authors:**

1. If the task is to unify video and text modalities, why is CLIP, not any video-text pre-trained models being used? How is CLIP used here to extract video embedding?

2. As you pointed out, LN184 "For Ego4D, a subset of the clips are also annotated with text narrations", how does the pre-training run differently on examples with and without text narrations?

**Reasons To Accept:**

Alignment across IMU and other modalities is relatively unexplored before and the proposed approach achieves good results in the main benchmarks, verifying the feasibility of unifying these modalities.

**Reasons To Reject:**

Although the idea about aligning more different modalities is important, the proposed approach is lacking bit in novelty. It is unclear what specific challenges are being solved in the work, on top of existing contrastive learning frameworks. Especially I feel IMU input modality may only be considered as a small set of concepts (e.g., say movements) that can be covered by video and text. Results from Table-2 also verified that video-text alignment is already done in good by CLIP, now how would it compare, e.g., if you do IMU->video retrieval followed by plain video->text, to that IMU->text in Table-2?

The definition of pre-training can be made more clear here. Which datasets are used for pre-training. LN203 “… evaluate on the held-out test set … “, seems to say both pre-training and the evaluation are actually on the same datasets.


**Reproducibility:**

4: Could mostly reproduce the results, but there may be some variation because of sample variance or minor variations in their interpretation of the protocol or method.

**Reviewer Confidence:**

3: Pretty sure, but there's a chance I missed something. Although I have a good feel for this area in general, I did not carefully check the paper's details, e.g., the math, experimental design, or novelty.

**Typos Grammar Style And Presentation Improvements:**

LN210 Task3 seems to be kept out as a future work. Maybe it suits better in other sections instead of Experiments.

---

> ### Author Rebuttal · Authors · 2023-08-29
>
> We thank the reviewer for their constructive comments and suggestions. We are appreciative of their acknowledgement of the novelty of the application, and the strong performance on downstream tasks.
>
> **[Re: “It is unclear what specific challenges are being solved in the work”]**
>
> This work addresses two crucial aspects for wearable device AI:
>
> - User privacy: Video recording raises privacy concerns as it's often intrusive and unavailable in sensitive situations, whereas IMU data remains non-intrusive.
> - Power consumption: In comparison to video recording, IMU signals are lightweight, resulting in significantly lower power consumption at inference time.
>
> Our exploration encompasses:
> - The design of an effective IMU encoder architecture that demonstrates strong performance in downstream tasks. Our ablation study (Table 5) shows that the proposed architecture outperforms other baseline architectures from the literature.
> - The alignment of IMU data within a shared language-vision space.
> - Pioneering applications based on IMU-LLM, showcasing reasoning capabilities while upholding privacy preservation.
>
> **[Cascaded retrieval.]**
>
> The reviewer proposes comparison of our IMU → Text retrieval task with a “cascaded retrieval” approach of IMU → Video → Text.
>
> While we agree that it is an interesting approach to consider, the approach lacks practical merits in the context of Wearable on-device AI, where lightweight power computation level & privacy are of utmost consideration (as opposed to cloud-based computations).
>
> - The cascaded approach assumes availability of a pool of videos as retrieval targets, which faces constraints such as (1) the memory limit of a wearable device, (2) user privacy considerations (around storing and accessing visual information).
> - The computation of vision embeddings typically requires much higher power and memory consumption levels than computing IMU embeddings.
>
> We will add more discussions and considerations on Wearable AI in the final manuscript as above, and also include new results comparing the cascaded approach.
>
> **[CLIP vs. Other Video-Text Pre-trained Encoders.]**
>
> We base our rationale for using CLIP for getting video representations as follows:
> - We focus on using the lightest model possible to encode vision signals at inference time, to match the computational budget we have for on-device use cases of our model.
> - [1] reports competitive performance with the CLIP-based approach using just a single frame, compared to several strong video-text models.
>
> In the final manuscript, we will include discussions on different choices of video encoders in terms of pre-training performance, and their respective computational requirement.
>
> [1] Portillo-Quintero et al., "A straightforward framework for video retrieval using clip", 2021.
> (https://arxiv.org/pdf/2102.12443.pdf)
>
> **[Pre-training schema.]**
>
> We split each dataset for upstream (alignment training) and downstream tasks (activity recognition). For IMU training, we only use the subset of the Ego4D data (540 hours out of 3,670 hours) that have text narrations, IMU and video alignment.
>
> **[Question Answering on Sensor Logs.]**
> Thank you for your suggestions. We will move this discussion to a new section, and add more qualitative analysis, page limit permitting.

---

### Official Review · Reviewer_JXWg · 2023-08-03

**Soundness:** 4

**Excitement:**

4: Strong: This paper deepens the understanding of some phenomenon or lowers the barriers to an existing research direction.

**Paper Topic And Main Contributions:**

This work presents a pre-training approach to align IMU sensor, video and text modality present in video data. The author optimizes contrastive loss between feature pairs from different modality, similar to the one used to pre-train CLIP, to align different modalities in a shared space. To show the effectiveness of their pre-training scheme in aligning representation from different modalities, the authors benchmark different cross-modal retrieval task.

**Reasons To Accept:**

1. From an application perspective, aligning text, video and IMU signal is interesting and is relatively unexplored.
2. Motivates using a lighter alternate to video modality for text-based cross-modal task.

**Reasons To Reject:**

1. Pre-training method is not novel, its an extension of contrastive feature loss extended to three different modalities.

**Reproducibility:**

4: Could mostly reproduce the results, but there may be some variation because of sample variance or minor variations in their interpretation of the protocol or method.

**Reviewer Confidence:**

3: Pretty sure, but there's a chance I missed something. Although I have a good feel for this area in general, I did not carefully check the paper's details, e.g., the math, experimental design, or novelty.

---

> ### Author Rebuttal · Authors · 2023-08-29
>
> We thank the reviewer for their constructive comments and suggestions. We are appreciative of their acknowledgement of the novelty of the application, and the strong performance on downstream tasks.
>
> **[Novelty of the proposed approach.]**
>
> We would like to highlight that the innovation within this study resides in the investigation of aligning a novel modality (IMU) through contrastive learning. Our exploration encompasses:
> - The design of an effective IMU encoder architecture that demonstrates strong performance in downstream tasks. Our ablation study (Table 5) shows that the proposed architecture outperforms other baseline architectures from the literature.
> - The alignment of IMU data within a shared language-vision space.
> - Pioneering applications based on IMU-LLM, showcasing reasoning capabilities while upholding privacy preservation.

---

### Official Review · Reviewer_X66w · 2023-08-05

**Typos Grammar Style And Presentation Improvements:** N/A
**Soundness:** 3

**Excitement:**

3: Ambivalent: It has merits (e.g., it reports state-of-the-art results, the idea is nice), but there are key weaknesses (e.g., it describes incremental work), and it can significantly benefit from another round of revision. However, I won't object to accepting it if my co-reviewers champion it.

**Missing References:**

N/A

**Paper Topic And Main Contributions:**

This paper proposes IMU2CLIP, where they pre-train a text/video-aware motion representation. Specifically, this paper utilizes contrastive loss to align the motion from Inertial Measurement Unit to text and video data. Empirical results show that the IMU2CLIP representation can be used for media search with low resources. Besides, fine-tuning with IMU2CLIP significantly outperforms baseline models trained from scratch on activity recognition tasks.

**Questions For The Authors:**

N/A

**Reasons To Accept:**

1. Interesting idea to connect motion from IMU with text and video representations.
2. Propose a low-resource way for media search.
3. Demonstrate strong performance on activity recognition when utilizing the pre-trained IMU2CLIP representation.

**Reasons To Reject:**

1. IMU sensor information has been used in multiple other motion recognition tasks. It's underexplored whether the proposed IMU2CLIP can achieve better performance on other established benchmarks.

**Reproducibility:**

4: Could mostly reproduce the results, but there may be some variation because of sample variance or minor variations in their interpretation of the protocol or method.

**Reviewer Confidence:**

3: Pretty sure, but there's a chance I missed something. Although I have a good feel for this area in general, I did not carefully check the paper's details, e.g., the math, experimental design, or novelty.

---

> ### Author Rebuttal · Authors · 2023-08-29
>
> We thank the reviewer for their constructive comments and suggestions. We are appreciative of their acknowledgement of the novelty of the approach and its new application, and the strong performance on activity recognition.
>
> **[Exploration of other benchmarks.]**
>
> We note that we report results on two datasets: Ego4D and Aria, which are the biggest & the latest datasets comprising IMU motion sensors (670+ hours total), covering diverse and unconstrained motions of multiple wearers (massive, unprecedented scale). On the other hand, the existing datasets are typically an order of magnitude smaller in size (<2 hrs of recording) and focused on constrained motion types. We therefore believe that the benchmark results we report on Ego4D and Aria can give a more generalized view on the performance than results on any other existing IMU datasets. To measure the performance improvement on focused studies, however, we will include results on the following publicly available IMU activity datasets [1-4] as well, in addition to Ego4D and Aria.
>
> - Ego4D: 923 participants, 540 hours, unconstrained motions
> - Aria: 138 hours
> - [1]: 8 participants, ~2 hours
> - [2]: 17 participants, ~2 hours
> - [3]: 114 participants, ~0.5 hour
> - [4]: 35 participants, ~1 hour
>
> [1] Kasebzadeh, Parinaz, et al. "IMU dataset for motion and device mode classification." 2017 International Conference on Indoor Positioning and Indoor Navigation (IPIN). IEEE, 2017.
> [2] Casilari, Eduardo, Jose A. Santoyo-Ramón, and Jose M. Cano-García. "Umafall: A multisensor dataset for the research on automatic fall detection." Procedia Computer Science 110 (2017): 32-39.
> [3] Nadeem, Adnan, Amir Mehmood, and Kashif Rizwan. "A dataset build using wearable inertial measurement and ECG sensors for activity recognition, fall detection and basic heart anomaly detection system." Data in brief 27 (2019): 104717.
> [4] Ribeiro De Souza, Caroline, et al. "A public data set of videos, inertial measurement unit, and clinical scales of freezing of gait in individuals with parkinson's disease during a turning-in-place task." Frontiers in Neuroscience 16 (2022): 832463.

---

### Meta-Review · Area_Chair_sv8f · 2023-09-19

**Recommendation:** 4

**Metareview:**

The paper proposes a method to learn to map wearable IMU data to the same space as text and video data, and shows that the method can be used for retrieval tasks across text, IMU and video modalities.

Strengths:
- All reviewers highlighted that aligning IMU with other modalities is an interesting idea and not explored before.
- Most reviewers commended strong results on the benchmarks, which show that the method works.
- Most reviewers agree that the method is a low-resource way to do media search (the wearable device can be low-power and low bandwidth).

Weaknesses that may have been addressed in rebuttal:
- One reviewer cited lack of comparison to prior motion datasets. The authors mentioned that this was because prior datasets are too small (<2 hours activity data), but they will run the experiments and include these results anyway at the request of the reviewer.
- R b6Gx raised a number of concerns and did not elaborate which were addressed during rebuttal. From my reading, it seems the concerns are addressed.

Overall, there is consensus that the paper studies an interesting and novel idea, and proposes a method that works, as substantiated by the experiments.

Overall: Sound and moderately exciting.

---

### Decision · Program_Chairs · 2023-10-07

**Decision:**

Accept-Findings

**Comment:**

The paper proposes a method to learn to map wearable IMU data to the same space as text and video data, and shows that the method can be used for retrieval tasks across text, IMU and video modalities.

Strengths:
- All reviewers highlighted that aligning IMU with other modalities is an interesting idea and not explored before.
- Most reviewers commended strong results on the benchmarks, which show that the method works.
- Most reviewers agree that the method is a low-resource way to do media search (the wearable device can be low-power and low bandwidth).

Weaknesses that may have been addressed in rebuttal:
- One reviewer cited lack of comparison to prior motion datasets. The authors mentioned that this was because prior datasets are too small (<2 hours activity data), but they will run the experiments and include these results anyway at the request of the reviewer.
- R b6Gx raised a number of concerns and did not elaborate which were addressed during rebuttal. From my reading, it seems the concerns are addressed.

Overall, there is consensus that the paper studies an interesting and novel idea, and proposes a method that works, as substantiated by the experiments.

Overall: Sound and moderately exciting.